biomechanics, evolution, palaeontology

axial skeleton, cervical column, evolution, Aves, ecological variation

**Author for correspondence:**
Ryan D. Marek
e-mail: rdmarek@liverpool.ac.uk

# Evolutionary versatility of the avian neck

Ryan D. Marek[1], Peter L. Falkingham[2], Roger B. J. Benson[3],
James D. Gardiner[1], Thomas W. Maddox[1] and Karl T. Bates[1]

[1]Department of Musculoskeletal & Ageing Science, University of Liverpool, William Henry Duncan Building, 6 West Derby Street, Liverpool, L7 8TX, UK
[2]Biological and Environmental Sciences, James Parsons Building, Byrom Street, Liverpool L3 3AF, UK
[3]Department of Earth Sciences, University of Oxford, South Parks Road, Oxford OX1 3AN, UK

RDM, 0000-0001-8139-4985; PLF, 0000-0003-1856-8377; JDG, 0000-0003-1902-3416; KTB, 0000-0002-0048-141X

Bird necks display unparalleled levels of morphological diversity compared to other vertebrates, yet it is unclear what factors have structured this variation. Using three-dimensional geometric morphometrics and multivariate statistics, we show that the avian cervical column is a hierarchical morpho-functional appendage, with varying magnitudes of ecologically driven osteological variation at different scales of organization. Contrary to expectations given the widely varying ecological functions of necks in different species, we find that regional modularity of the avian neck is highly conserved, with an overall structural blueprint that is significantly altered only by the most mechanically demanding ecological functions. Nevertheless, the morphologies of vertebrae within subregions of the neck show more prominent signals of adaptation to ecological pressures. We also find that both neck length allometry and the nature of neck elongation in birds are different from other vertebrates. In contrast with mammals, neck length scales isometrically with head mass and, contrary to previous work, we show that neck elongation in birds is achieved predominantly by increasing vertebral lengths rather than counts. Birds therefore possess a cervical spine that may be unique in its versatility among extant vertebrates, one that, since the origin of flight, has adapted to function as a surrogate forelimb in varied ecological niches.

## 1. Introduction

How, why and at what scale phenotypic variation arises in morphological structures are among the most important questions in evolutionary biology [1]. The avian neck is a highly modular structure [2–9] that displays a wide array of morphological diversity. As the forelimbs are dedicated to flight, the neck has adopted the role of aiding the beak in environmental manipulation tasks [6,10,11]. Phenotypic variation appears in many forms throughout the avian neck; counts of cervical vertebrae vary between 10 and 26 [12,13], as opposed to the count of seven found in almost all mammals [14–16]. Avian vertebral morphologies [3–10] and overall neck length [11] also display a wide diversity of form. However, no previous body of work has quantitatively addressed the ecomorphological signal in this variation, despite the clear functional significance and variability of the avian neck [2,17,18].

Previous work on the vertebrate neck has provided insights into key innovations and other traits that were pivotal to the diversification of major groups, such as neck elongation in sauropod dinosaurs [19], neck retraction mechanisms in turtles [20] and patterns of vertebral fusion across tetrapods [21]. However, much previous investigation into the drivers of phenotypic variation of cervical morphology has focused upon mammals [16,22–24], which possess developmental and genetic restrictions on counts of cervical vertebrae [25,26], potentially limiting the capacity for functional variation. By contrast, avian necks show great capacity for evolutionary variation [3–15], but the effects of

ecological adaptation and intrinsic constraints on avian neck evolution have not been quantified. This represents a major gap in understanding of the phenotypic variation of the vertebrate neck and means that the role that neck has played in the evolution of birds into one of the most taxonomically, morphologically and ecologically diverse groups of vertebrates is poorly constrained.

A key unanswered question concerns the extent to which phenotypic variation of the avian cervical column is driven by adaptive responses to extrinsic (ecological) factors, or by intrinsic (scaling) constraints. This question has not been systematically addressed, due in part to continuing disagreement about how the avian cervical column should be compartmentalized into sub-regions [2,5,6,8–10,27] and subsequently compared across species. Over the past 90 years, the avian neck has been sub-divided into between 3 to 9 regions by different workers based on disparate methods, such as variation in joint motion [8,9], qualitative comparative anatomy [3,6,27,28] and quantitative shape analysis techniques [2,10]. Crucially, in all these cases the link between regionalization and the genetic and development homology of the neck remains unclear. The boundaries between axial regions in vertebrates are delineated by *Hox* gene expression limits [10,29]. Recent work has shown that these expression limits delineate five morphological regions within the cervical column of *Gallus gallus domesticus* [10], raising the possibility that these five regions may be homologous across all extant Aves. Issues concerning homology of individual vertebrate among species with differing cervical counts could potentially be resolved using quantitative information on morphological similarities to define regions, as a proxy of *Hox* gene expression limits [10]. However, the hypothesis that five cervical regions are present across extant Aves has not been directly tested with a broad comparative sample.

Here, we apply a combination of three-dimensional geometric morphometrics and phylogenetic comparative methods to the cervical column of a diverse array of extant birds to investigate morphological variation at multiple scales (whole-neck, regional and sub-regional) and its association with key intrinsic (body size, neck length, head mass) and extrinsic (diet, locomotion) factors. Our analysis recovers five morphological sub-regions, consistent with *Hox* gene expression limits [10], in representative birds from all major extant taxonomic sub-groups, locomotor and tropic ecologies. Analysis of this homologous five-region structure highlights a highly scale-dependent nature of phenotypic variation in the avian neck, with varying degrees of ecological adaptation at macro- to micro-morphological scales. To our knowledge, this is the first quantitative demonstration of hierarchical ecologically driven morphological organization of the vertebrate neck and suggests that similar assessments of anatomical variation at different scales could provide important insight into diversity and adaptation in the necks of other amniote groups and indeed across the vertebrate skeleton.

## 2. Methods

### (a) Specimen digitization and assessment of regionalization

Three-dimensional digital models were created for every cervical vertebrae (except the atlas, C1) for 54 specimens (48 distinct species, electronic supplementary material, table S2) of extant birds, from medical and microCT scans using Avizo 7.1 (Visualisation Science Group). To characterize vertebral morphology, we used the combination of 15 morphological landmarks and qualitative characters shown previously to delineate morphological regions that are consistent with *Hox* gene expression limits in *Gallus gallus domesticus* [10] (electronic supplementary material, figure S1 and table S1). For each individual bird, landmarked vertebrae were subjected to a generalized Procrustes analysis to remove the effects of size and rotation using MorphoJ. A suite of qualitative characters was recorded for each species, which recorded vertebral shape change along the length of the cervical column that was not accounted for by the landmark scheme [10]. The Procrustes coordinates were then combined with the qualitative characters of that species to produce a principal coordinates analysis (PCA) plot and cluster dendrogram via a Gower single-linkage algorithm in PAST 3.0 [30]. Regionalization was then assessed based on groups of vertebrae that clustered together on the dendrogram which displayed smaller distance measures with each other than to other vertebrae, along with bootstrap values for that node. When support values were low candidate homologies based on the similarity of form were assessed from the PCA plots [10].

### (b) Explanatory variables

Head mass was quantified digitally by using an α-shape fitting algorithm [31] on three-dimensional models of CT scanned skulls of all 38 species. α-shapes were fitted to skulls using an in-house modified version of the 'alphavol' package in MatLab, which calculates the volume of the computed α-shape. Head mass was estimated by multiplying the α-shape volume by the weighted mean density of soft tissues within the skull (approximated to the density of water, $997 \text{ kg m}^3$) and normalized by taking the head mass as a percentage of total body mass. Body masses were weighed directly when possible. When not possible, body masses were estimated using scaling equations based on femoral length, minimal circumference of the femoral shaft and humeral articulation facet on the coracoid [32], and an average was taken. Neck length was measured digitally as the summed length of each individual cervical vertebrae of each bird. Neck length was normalized by neck length/(body mass$^{0.33}$). Diets were assigned to birds based on data from multiple volumes of *Handbook of the Birds of the World* [33–43], and locomotor mode was collated from the literature [44–46].

### (c) Phenotypic trajectory analysis

A dataset consisting of the mean vertebral shape for each of the cervical regions for all birds in the study was created from the results of the PCA and cluster analysis. This dataset was then subject to an initial generalized Procrustes analysis and PCA within the R package 'geomorph' [47]. Phenotypic trajectory analysis (PTA) [48] was used to quantify ecological and phylogenetic effects on shape across the entire cervical column. In this instance, PTA plots a trajectory through shape space for a specific group within a factor (flightless birds within locomotor ecology, for example) by connecting the mean shape for each cervical region for that specific group with the mean shape of the next region (from region 1 to region 2, then from region 2 to region 3 etc.) until all cervical regions are connected and form a trajectory that represents the shape change across the entire cervical column.

### (d) Procrustes distance generalized least-squares modelling

Procrustes distance phylogenetic generalized least-squares (D-PGLS) was used to model relationships between mean regional vertebral shape and extrinsic factors (dietary ecology,

locomotory ecology, head mass, body mass and neck length). This, and other phylogenetic comparative methods, used a distribution of supertree topologies from previous analyses [49]. There is no AIC framework for D-PGLS. Therefore, models were compared based on rankings of residual $R^2$. The best model was determined to be the model with the lowest residual $R^2$ whereby all factors included in the model had a significant $p$-value (less than 0.05). Redundancies among pairs of variables were evaluated by consideration of their $R^2$ and $p$-values in models that contained both variables together.

### (e) Neck length allometry

Data for head mass, neck length and body mass could be collected from 38 out of 52 specimens used throughout the previous sections. Phylogenetic generalized least-squares (PGLS) regression [50] was used to model scaling relationships between neck length, head mass and body mass in a phylogenetic framework in R using 'nlme' and 'ape' packages. Pagel's $\lambda$ [51] was used with a freely varying parameter to assess the impact of phylogeny on statistical models and to scale them accordingly. Models were compared based upon rankings of AICc scores.

### (f) Phylogenetic ANOVA of region lengths and counts

D-PGLS (in the R package 'geomorph' [47]) was used to assess the correlation between regional counts of cervical vertebrae and intrinsic (size) and extrinsic factors (diet, locomotion). The coefficients of these relationships were examined to assess the effect of each factor on region size in each region. Region length was measured digitally as the summed length of each individual cervical vertebrae for each of the five regions for each specimen. D-PGLS was used to model relationships between region length (coded as a multivariate factor) and extrinsic factors in a similar manner to methods presented in 'Procrustes distance generalized least-squares modelling' above. The coefficients of these models were used to observe the effect of each factor on region length for each region. Models were compared based on rankings of residual $R^2$.

## 3. Results

### (a) Conservatism in avian cervical regionalization

Five cervical regions can be identified across all species in the dataset using PCA and cluster analysis (figure 1a; see Methods; electronic supplementary material, table S1). Regions 2–5 all display considerable variation in both vertebral counts and region lengths (figure 1a). PCA morphospace occupation of each region was conserved for each individual bird studied, with each region occupying a distinct area of morphospace in all species, with regions 3–5 displaying some overlap (figure 1b,d). Region 1 (always consisting of just C2) occupies the most distinct region of morphospace when all birds are considered (figure 1d), and regions 2 and 5 also occupy distinct areas of morphospace, albeit to a lesser extent. The large overlap in morphospace occupation indicates that the third and fourth regions are more morphologically similar to each other (figure 1d). Comparisons of mean region shape data (PCA plots, figure 1d) across species reveals that a common pattern of shape change along the cervical column exists among all extant birds and that each cervical region has identifiable features of vertebral anatomy (figure 1c).

### (b) Inter-regional cervical morphology correlates only with specialized extrinsic factors

PTA allows pairwise comparisons of inter-regional vertebral morphology for both ecological factors (diet, locomotion) and taxonomic groups [52,53] (figure 2). Diet has little correlation with shape variation across the entire cervical column, with only carnivores and insectivores recovered as significantly different from each other in trajectory direction and shape ($p = 0.025$ and $p = 0.025$ respectively, figure 2a–c). Carnivores have a relatively enlarged, more upright neural spine of region 1 (figure 2d), whereas regions 2 and 3 are similar between carnivores and insectivores (figure 2d,e). Insectivores have a shallower neural spine in region 4 when compared to carnivores (figure 2d,e), while carnivores display more variation in centrum length and height between regions 3 and 4 (figure 2d,e). This pattern is also observed between regions 4 and 5 of insectivores (figure 2d,e).

Only two locomotor groups (soaring and continual flapping flight) showed statistically significant differences in the PTA, and this difference was restricted to trajectory direction ($p = 0.045$, figure 2b,f,g). Continual flappers have a shorter neural spine than soaring birds and display less inter-regional variation between regions 3 and 4 (figure 2f, g). Some features of regional morphological variation are specific to comparisons of soarers and continual flappers, with the angle of orientation of the prezygopohyseal articular facet changing to a greater degree between all 5 regions, as well as the cranocaudal enlargement of the neural spine of region 1 in soarers (figure 2f,g). Taxonomic groupings displayed no significant differences between all three trajectory descriptors (figure 2f,g).

### (c) Finer-scale (intra-regional) cervical morphology correlates with extrinsic factors

We used D-PGLS to model the effect of both intrinsic (body mass, neck length, head mass) and extrinsic (diet, locomotion) factors on the mean shape of each individual cervical region while accounting for shared evolutionary history (electronic supplementary material, table S3, tables S6–10). Models were then compared based upon minimal residual $R^2$ rankings (see Methods for more information). Here, we summarize the significant findings for each cervical region. More detailed description of these results can be found in the electronic supplementary material.

Vertebral shape in regions 1 and 2 are best explained by a combination of neck length and intermittent bounding (electronic supplementary material, table S3, S6). In region 1 intermittent bounding has considerably greater $R^2$ than neck length, while this is reversed in region 2. In region 1, intermittent bounding is associated with a tall neural arch and a cranially shifted centrum (electronic supplementary material, figure S3). Increases in neck length are associated with an elongation of the centrum and a flattening of the neural spine in region 2 (electronic supplementary material, figure S2a,b). Intermittent bounding is associated with a heightened neural spine and a flattened centrum in region 2 (electronic supplementary material, figure S3). Vertebral shape in region 3 is best explained by a model containing neck length, flightlessness and carnivory (electronic supplementary material, table S3, S8). Within this model, flightlessness displayed the highest value of $R^2$ and this ecology is associated

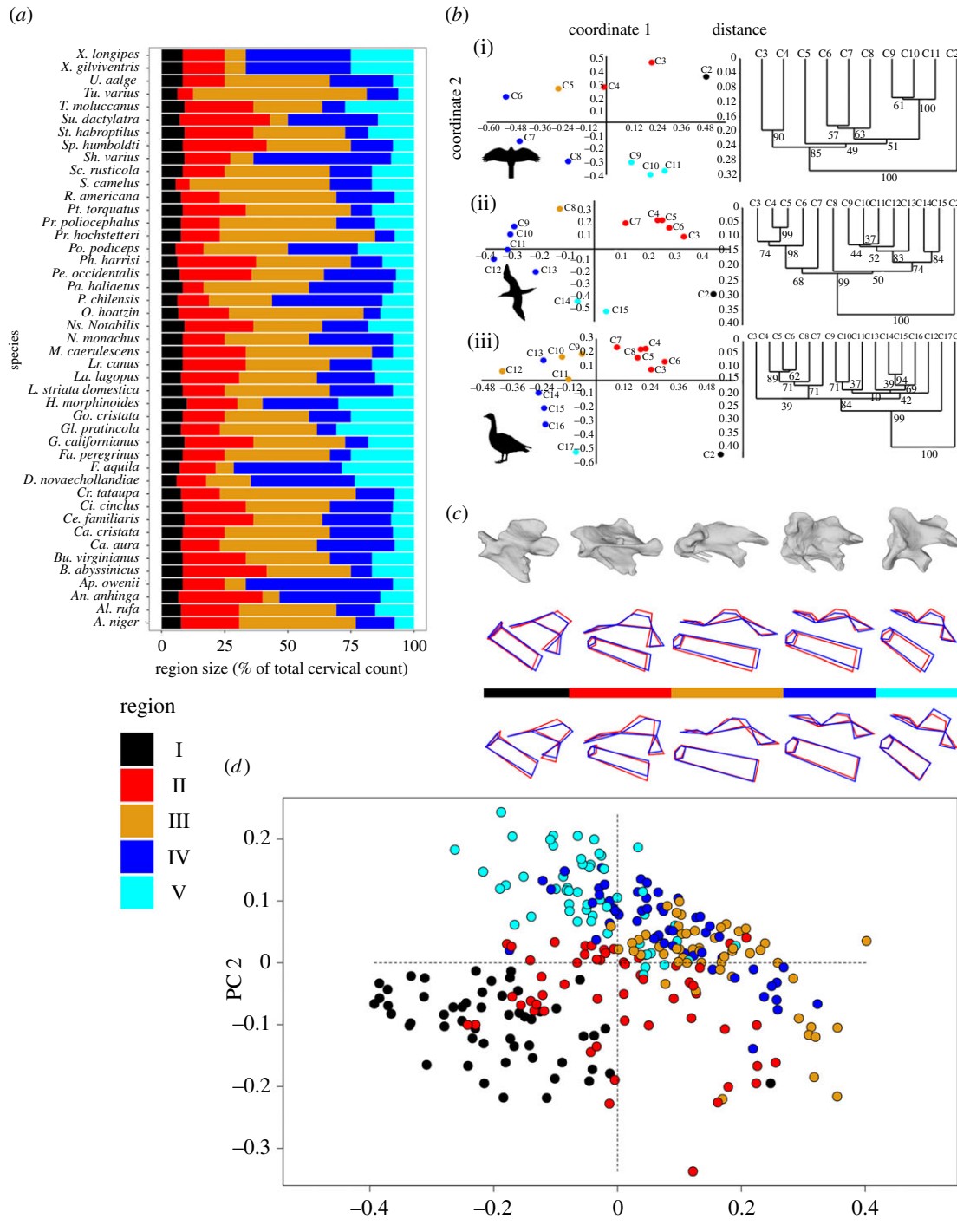

**Figure 1.** Conservatism in avian cervical regionalization. (*a*) Region size (as a normalized measure of percentage of total cervical vertebrae) variation for all birds studied. Colours denote region number. All extant birds have five cervical regions. Regions 1 and 5 are stable in their number of vertebrae, whereas regions 3 and 4 display the largest variations in vertebrae per region. (*b*) Principal coordinate graphs (left—proportions of variance explained by each axis on the axis labels) and cluster analysis charts (right) depicting the delineations between cervical regions in three taxa: (i) *Hieraaetus morphnoides*, (ii) *Sula dactylatra* and (iii) *Branta leucopsis*. Colours on principal coordinate graphs denote cervical regions. Numerical values underneath cluster branches denote bootstrap support after 1000 replicates. Despite changes to total number of cervical vertebrae and ecology, all birds display five cervical regions when PCA and cluster analyses are used together to designate regions. (*c*) Upper: shape change across PC1 for all five regions (the colour bar indicates region number, cranial regions are on the left), with CT images of vertebrae from each region above (species: *Alectoris rufa*). Red outline denotes mean shape, blue outline displays the maximum shape change across PC1. Lower: shape change across PC2 for all five regions (colour notations are as in (*b*)). Region 1 is defined by an anteriocaudally restricted centrum length, a deepened centrum, a tall neural spine, and small prezygopophyses with cranially facing articular facets. Region 2 retains the enlarged neural spine but displays an elongated, thinner centrum and larger more cranially positioned prezygopophyses. Region 3 displays the smallest neural spine of all five regions, as well as the most elongate centrum, with the articular facets of the prezygopophyses facing dorsocranially, while the facet of the postzygapophyses are oriented ventrocaudally. Neural spine height increases slightly within region 4, while the centrum is shorter and deeper than in region 3, and the articular facets of the prezygapophyses are more dorsally oriented in region 4. Region 5 displays a larger neural spine still, with a shorter and deeper centrum, the articular facet of the prezygapophyses face more cranially than in region 4. (*d*) PCA of mean regional vertebral shape for all birds. For all birds, shape change along the first principal component involves a variation in the height of the neural spine, rotation of both pre- and postzygophyseal articulation facets, and a cranocaudal variation in the length of the centrum. (Online version in colour.)

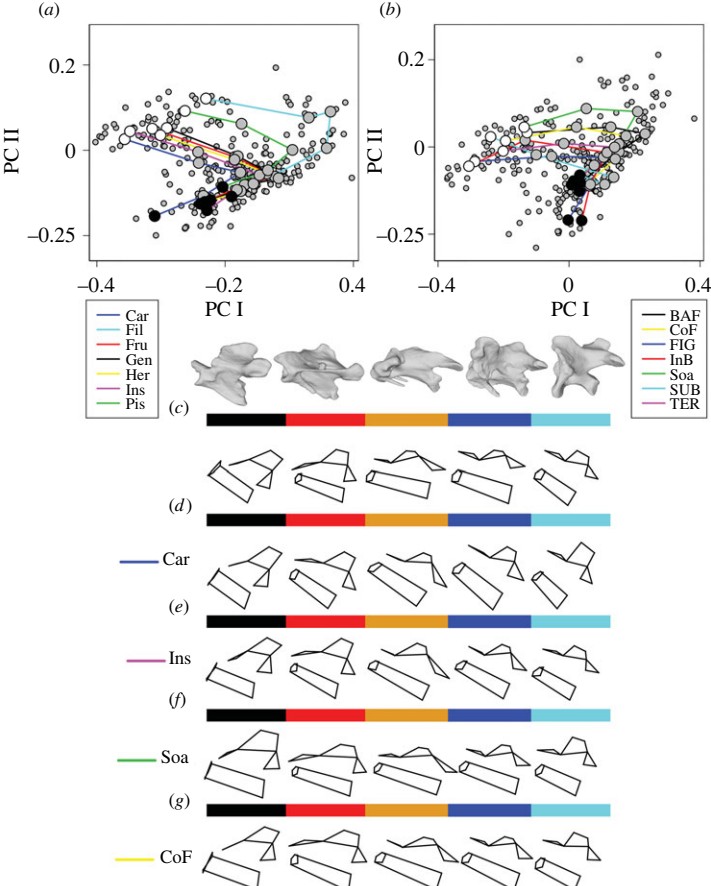

**Figure 2.** Inter-regional cervical morphology correlates only with extrinsic factors that have specialized cervical kinematics. (*a*) Phenotypic trajectories of dietary ecologies. Colours denote diet. Car = carnivory, Fil = filter feeding, Fru = frugivore, Gen = generalist, Her = herbivore, Ins = insectivore, Pis = piscivore. PTA analyses shows that despite large differences in dietary ecology, the gross morphology of the entire cervical column does not change (except between ecologies that are extremely divergent, see electronic supplementary material, table S5 below). Black circles represent the group mean region shape for region 1, white circles represent group mean region shape for region 5. Grey circles represent the group mean region shape for regions 2–4. (*b*) Phenotypic trajectories of different flight styles. Colours denote flight. BAF = burst-adapted flight, CoF = continual flapping, FIG = flap gliding, InB = intermittent bounding, Soa = soaring, SUB = subaqueous, TER = terrestrial. As for dietary ecologies, locomotory mode (flight style) has little impact on gross morphology across the entire cervical column (except in extremely divergent taxa, see electronic supplementary material, table S5 above). (*c–g*) Mean regional shape change across the cervical column with CT images of vertebrae from each region above (species: *Alectoris rufa*). Lateral view of each region mean vertebral shape, colour indicates region, cranial regions are towards the left, caudal regions are to the right. (*c*) Mean region shapes for all birds, (*d*) mean region shapes for carnivorous birds, (*e*) mean region shapes for insectivorous birds, (*f*) mean region shapes for soaring birds and (*g*) mean region shapes for continual flapping birds. (Online version in colour.)

with an elongated centrum and an increased height of the neural spine (electronic supplementary material, figure S3). Increases in neck length are associated with an elongation of the centrum and a flattening of the neural spine (electronic supplementary material, figure S2a, b). Vertebral shape variation in region 4 is best explained by a model that contains neck length, flightlessness, intermittent bounding and carnivory (electronic supplementary material, table S3, S9). Within this model, carnivory possesses the highest $R^2$ value (electronic supplementary material, table S3, S9). Increases in neck length are associated with a deepening and elongation of the centrum as well as an increase in neural spine height (electronic supplementary material, figure S2a, b), while carnivory is associated with a shortened centrum and an increase in neural spine width and height in region 4 (figure 2d). The highest-ranking model (electronic supplementary material, table S3, S10) that explains vertebral shape in region 5 contains body mass, intermittent bounding and carnivory. Carnivory displays the highest value of $R^2$ within this model (electronic supplementary material, table S3, S10).

Increases in body mass are associated with an increase in robusticity and height of the neural spine and centra (electronic supplementary material, figure S2a,b). Carnivory is associated with a dramatic increase to the width and height of the neural spine, which is also angled more cranially compared to the species average (figure 2c,d).

## (d) Isometric scaling of neck length in birds but with considerable variability

pGLS models recover statistically significant isometric relationships between neck length and body mass (electronic supplementary material, figure S4a) and head mass (electronic supplementary material, figure S4b) in our sample of birds, but with considerable scatter in the data (electronic supplementary material, figure S4, tables S4, S11). Model comparisons reveal that neck length variation is best explained by a model that contains head mass only (AICc = −6.90; electronic supplementary material, table S4). The coefficient of head mass in this model indicates isometric scaling of head

mass with neck length (coefficient = 0.319, CI = 0.066; electronic supplementary material, tables S4, S11; figure S4*b*), and Pagel's $\lambda$ indicates strong phylogenetic signal ($\lambda = 1.009$; electronic supplementary material, tables S4, S11). A model that also includes insectivory is also relatively well-supported (AICc = −5.02; electronic supplementary material, tables S4, S11) and has a marginally significant coefficient indicating that insectivorous birds have a generally longer neck length than other birds in the study (coefficient = 0.141, SE = 0.068). The relationship between neck length and body mass becomes non-significant when included in a model with head mass. The same is true for flightlessness (electronic supplementary material, tables S4, S11).

### (e) Mechanisms of avian neck elongation

D-PGLS was used to model the relationships between region lengths and regional vertebral counts with intrinsic scaling factors and extrinsic ecological parameters. Model comparisons were used to assess which models best explained variation in the data. Region lengths are best explained by a combination of neck length and soaring (residual $R^2 = 0.5676$; electronic supplementary material, tables S5, S12). Within this model, neck length has more explanatory power than soaring does ($R^2_{necklength} = 0.3031$, $R^2_{soaring} = 0.1374$; electronic supplementary material, tables S5, S12). The coefficients of the individual regions from the region lengths~neck length model reveal that neck elongation is primarily achieved by increases to the lengths of vertebrae in regions 2–4, and especially by increases in region 3 (region 3 coefficient = 8.6715; electronic supplementary material, tables S5, S12). Soaring birds appear to display a strong decrease in the lengths of vertebrae in region 3, which is accounted for by a sharp increase of lengths in region 4 (region 3 coefficient = −6.9142, region 4 coefficient = 6.4572; electronic supplementary material, tables S5, S12). Models that contain body mass and head mass separately are significant but are less supported than models containing neck length (residual $R^2_{bodymass} = 0.8272$, residual $R^2_{headmass} = 0.8318$; electronic supplementary material, tables S5, S12).

Region counts were best explained by soaring alone (residual $R^2 = 0.8884$; electronic supplementary material, tables S5, S13). Coefficients reveal that vertebral counts are decreased in the third region of soaring birds and this is accounted for by an increase in counts to region 4 (region 3 coefficient = −2.1939, region 4 coefficient = 2.1990; electronic supplementary material, tables S5, S13). Frugivory also has a significant relationship with regional counts of vertebrae but this model is less supported, and when frugivory is combined with soaring in a single model, frugivory becomes redundant ($p = 0.07$; electronic supplementary material, tables S5, S13). No intrinsic scaling factors had a significant correlation with regional vertebral counts ($p = > 0.05$; electronic supplementary material, tables S5, S13); this indicates that neck elongation in Aves is not achieved by additions to vertebral counts.

## 4. Discussion

Our analyses highlight that the avian cervical column is a hierarchical morpho-functional appendage, with varying magnitudes of phenotypic variation at different scales. We find that patterns of shape variation across the entire neck as well as vertebral counts are not matched by high levels of variation in overall construction and regional modularity of neck. The phylogenetically broad and ecologically diverse sample of birds studied here all show five homologous regions (figure 1), characterized by a similar pattern of shape change between all regions (figure 2*a*,*b*) and few significant correlations between overall neck length, region lengths and ecology (electronic supplementary material, tables S4, S5, figure S2). Only mechanically demanding ecological behaviours appear to be associated with statistically significant modifications to this universal structural and morphological blueprint (figure 2*c*–*g*). Our results also reveal, contrary to previous expectations [5,6,54,55], that lengthening of vertebrae rather than cervicalization (the addition of vertebrae to the neck) drives neck elongation in birds, and that neck length scales isometrically with both body and head size (figure S4) with little ecological signal (electronic supplementary material, figure S2 and table S4). In spite of this overall conservation of neck architecture, our analyses of intra-regional osteological variation indicate that intrinsic and particularly extrinsic factors do exert significant adaptive morphological changes (electronic supplementary material, figure S2 and table S3), representing finer-scale modifications to the generalized avian cervical system.

### (a) The avian neck: a hierarchical morpho-functional structure

Our finding that birds of diverse taxonomic affinity, and varied locomotor and dietary ecology, share the same five cervical regions (figure 1) suggests that regional organization may be homologous across all extant Aves. By contrast, crocodilians and basal non-avian dinosaurs have been shown to possess only four cervical regions [11], and new analysis of two non-avian theropods (an allosauroid and a dromaeosaurid; electronic supplementary material, figure S5) also recovers four regions in these taxa. This new data suggests that the evolution of five cervical regions may be an avian-specific synapomorphy; however, further work investigating the regionalization of the theropod cervical column is needed to confirm this. The timing and selective pressures behind the evolution of a fifth cervical region in birds remain unknown, but it is tempting to speculate that increased regional differentiation in the neck may be causatively associated with expanded locomotor and dietary ecology in birds compared to their non-avian theropod ancestors. Indeed, previous work suggests that a shift away from carnivory may have facilitated shifts in cervical count in herbivorous theropods [56]. Alternatively, selective pressure exerted on the cervical system by the evolution of flight, as the neck took over from the forelimbs as the primary appendage for environmental manipulation, may have driven the evolution of the five-region system seen in modern birds.

PTA demonstrates that inter-regional morphology across this five-region cervical system is highly conserved across the majority of dietary and locomotor modes used by extant birds (figure 2). The neck of vertebrates primarily supports the head, providing it with a degree of movement and allowing the head to partake in a multitude of functional tasks (feeding, vigilance, conspecific interaction etc.) [57,58]. In this regard, the head and neck may act together to allow the head to be used as a 'hand' in order to interact with the environment in the stead of forelimbs that are primarily adapted for flight. Results from the PTA (figure 2) lend credence to this

hypothesis as patterns of morphological variation across the entire cervical spine as a whole are conserved across the majority of species studied, suggesting that these patterns may be adapted for providing the neck with generalized kinematics. Birds share patterns of cervical kinematics for many activities and the conservative nature of regionalization and inter-regional variation found herein provides the morphological evidence the avian neck, generally, may be adapted to the 'economics of continuous movement' than to any specific ecology or behaviour [4,6,54,55]. Alternatively, the retainment of consistent overall morphological blueprint across most ecological groups may represent constraints imposed a conserved pattern of *Hox* gene expression, although that modifications have evolved in response to mechanically demanding neck functions (figure 2) suggests it is most likely a product of both genetic and functional influences.

Within dietary categories, only carnivores and insectivores differ from each other in their macro-level (inter-regional) morphology (figure 2d,e). That these two particular ecologies are associated with larger-scale modular changes to the neck is consistent with the fact that they require 'extreme' and also juxtaposing mechanical demands: carnivory requires relatively slow but forceful retraction to tear flesh from prey [18] (a force- or work-based system retraction), while insectivory involves relatively high velocity protraction or extension of the neck to help capture escaping prey [59] (a velocity- or power-based system of extension). Carnivorous birds have a significant association with cervical morphology across both inter- and intra-regional analyses, and this effect is concentrated on regions 4 and 5 of the cervical spine (figure 2; electronic supplementary material, figure S2, table S3). Large retraction forces required during the 'pull' phase of feeding, as the flesh is torn from the prey [18,60,61], are generated by muscles of the M. longus colli dorsalis complex [61,62] which attach to the neural spines of vertebrae in regions 4 and 5. The increased height of the neural spines of vertebrae in these regions (figure 2d) may serve to increase the moment arm and attachment area of these muscles to power carnivorous cervical kinematics, as well as increasing stiffness at the base of the neck [63,64], to provide stability during the 'pull' feeding phase. Such adaptations to enhance the force and torque generating capacity of retractors muscles are likely to be relatively redundant in insectivores and indeed may actually be disadvantageous to both rapid neck extension and, in some taxa, rapid flight [65]. Recent work has suggested variation exists in the neck musculature of vultures [17], and it would therefore be interesting to examine finer-scale adaptations in the cervical system within groups such as carnivores (e.g. to assess potential adaptive responses to different prey types and modes of carnivory, such as predation versus scavenging).

Within locomotor categories, only soaring and continual flapping birds differ from each other in their macro-level (inter-regional) morphology (figure 2f,g). These two groups differ only in trajectory direction with the PTA analysis, suggesting their inter-regional disparity is less than seen between carnivores and insectivores, which differ in both trajectory direction and shape (figure 2). Vision must be stabilized during flight and this is achieved by oscillating movements of the neck and head that counteract each wing-beat [66,67]. Soaring birds flap less during a flight than continual flappers, and our results show they display more morphological differentiation (figure 2f) compared to the

all-birds model (figure 2c) than continual flappers do (figure 2g). This may be an indication that compensatory movements of the neck that stabilize vision during flight may constrain vertebral morphology to an extent.

While at the macro-scale, we find a conserved pattern of modularity and inter-regional morphology (figures 1 and 2), it is clear that phenotypic variation is present in the avian neck at the intra-regional scale and this diversity is correlated with intrinsic and extrinsic ecological factors. Indeed, our statistical analyses suggest that variation in intra-regional morphology correlates more widely with extrinsic than intrinsic factors (electronic supplementary material, table S3), which (to our knowledge) represents the first quantitative demonstration of ecologically associated morphological variation in the avian cervical column.

Neck length (intrinsic) and carnivory, flightlessness and intermittent bounding (extrinsic) are consistency correlated with intra-regional morphological variability in the neck (figure 2d; electronic supplementary material, table S3, figure S3), suggesting adaptive responses in osteology to these factors. These extrinsic ecological parameters appear to have the strongest correlation with vertebral morphological variation in regions 1 (intermittent bounding), 3 (flightlessness), and 4 and 5 (carnivory). Neck length shows the single strongest correlation in region 2. As in the case of carnivory, vertebral morphology associated with intermittent bounding and flightlessness also appear to represent logical adaptations to the mechanical demands placed on the neck in these behaviours. During flight, birds must stabilize their gaze in order to safely and efficiently navigate their flight path [68]. Intermittent bounding involves an active flapping phase followed by a passive phase whereby the wings are folded and the bird follows a ballistic trajectory, thus vertical oscillations are introduced into the flight path [69]. To maintain a stable gaze the head must be able to counteract these movements, and do so via dorsal head flexors such as M. complexus, M. rectus capitis dorsalis and M. longus colli dorsalis pars cranialis [66,67]. The neural spines of vertebrae in cervical regions 1 to 3 are heightened in intermittent bounders, and as many of these dorsal head flexors attach to this feature of vertebral anatomy [70], this may represent an adaptive response to counteracting the oscillations during bounding flight. Terrestrial locomotion requires the neck to stabilize vision while traversing variable terrain and at fluctuating speeds, and as such a combination of passive ligament support and active muscle force activation across the entire neck is required [71,72]. Flightless birds possess robust vertebral morphologies across the entire cervical spine and this may be an adaptation to providing a larger attachment area for multiple neck- and head-supporting to provide adequate vision stabilization during terrestrial locomotion. This hypothesis is supported in part by previous work finding that multiple species of flightless birds possess enlarged cervical muscles across the entire neck [6,73]. Flightless birds are also not constrained by selective pressures for weight reductions necessary for flight and this may at least partly explain their more robust cervical musculoskeletal system.

## (b) The unique nature of avian neck elongation, its constraints and variability

Avian neck length scales isometrically with head mass and body mass (electronic supplementary material, figure S4,

table S4). This differs from other groups of vertebrates (e.g. negative allometry in mammals [24]), as was also reported in other recent work [11]. Here, we show that increased vertebral length is the primary mechanism by which neck elongation occurs (electronic supplementary material, table S5), rather than through the addition of vertebrae as might be supposed given the high levels of variation in cervical vertebrae number across extant birds. In other groups of vertebrates (mammals), it is the weight of the head that is the predominant constraint upon neck length, as head mass scales at a faster rate than the cross-sectional area of the neck which must resist the stress of the weight of the head [74–78]. Our findings suggest that this constraint appears to be removed in birds, as both head mass and neck length scale isometrically with body mass and with each other (electronic supplementary material, table S4 and figure S2). Head mass is reduced in birds due to the negative scaling of the brain and eye size with body mass [79,80], the reduction of jaw musculature (as food processing occurs in the gizzard [81]) and the widespread pneumatization of the skull [82]. Morphological adaptations of vertebrae may also contribute to the release of constraint in neck length. Vertebrae in the mid-portion of the neck display multiple adaptations to increased intervertebral flexion in response to increasing neck length and these features allow the neck to achieve the 'S' shaped curvature seen across Aves [5]. This curvature allows the mass of the head to be held closer to the centre of mass, and in tandem with the lightweight head, this combination of craniocervical traits allows for a variety of head shapes and sizes to be supported by an elongated neck, overcoming the constraint of head mass that is present in many other vertebrates. Some additional discussion of the weak association between neck length and ecology is presented in the electronic supplementary material.

While isometric scaling of the neck with respect to body and head mass is recovered as statistically significant (electronic supplementary material, table S4), there is clearly considerable variation in the data and this is only partially explained by the intrinsic and extrinsic variables assessed here (electronic supplementary material, figure S4, tables S4, S11). Our analyses consistently recover strong phylogenetic signals in regression models, suggesting that phylogenetic history may explain a sizeable portion of the observed variation in cervical morphology. Variables not considered in this study may also explain at least some of the variability in our data. For example, a recent study recovered a relationship between neck length and leg length in birds, and suggested this relationship was a product of the need to maintain a neck length capable of allowing the head to reach the ground [11]. It is perhaps likely that variation in relative neck lengths we observe here (electronic supplementary material, figure S4) are reflective of the multiple intrinsic and extrinsic selective factors acting upon the avian neck, which must function as a multi-purpose surrogate arm (see previous section).

Cervicalization (increases to the number of cervical vertebrae) was previously thought to be responsible for neck elongation in birds [5,6,54,55]. However, our data provide no evidence in support of this hypothesis and instead suggest that vertebral elongation is the primary mechanism by which neck elongation occurs. Specifically it is increases in the length of vertebrae in all regions except region 1 that are the epicentres of neck elongation across Aves (electronic supplementary material, table S5). This is in contrast with the more localized method by which mammalian neck elongation occurs, as it is vertebrae from just the middle portion of the neck length that lengthen [24]. Our results therefore suggest that birds are not only unique in showing morphological responses to extrinsic ecological factors at multiple hierarchical levels, but also in their patterns of neck length allometry and elongation. Amalgamated, these results suggest that birds possess a cervical spine which is unique in its construction and elongation among vertebrates, and one that has adapted to the burden of becoming a surrogate forelimb as well specializing under ecological pressures.

Data accessibility. Data are provided as part of the supplementary information. Three-dimensional bones models are available from: https://doi.org/10.5061/dryad.0k6djhb02. A selection of models are also available on morphosource (see electronic supplementary material2.xlsx file for links).

Authors' contributions. R.D.M., K.T.B. and P.L.F. devised the project. R.D.M., R.B.J.B, T.M. and J.G. developed the methodologies. R.D.M., K.T.B and R.B.J.B collected the data. R.D.M. analysed the data and wrote the manuscript. All authors edited and reviewed the manuscript.

Competing interests. We declare we have no competing interests.

Funding. Funding for this project has been provided by a doctoral dissertation grant from the Adapting to the Challenges of a Changing Environment (ACCE) NERC doctoral training partnership (NE/S00713X/1) to R.D.M and K.T.B, and the European Union's Horizon 2020 research and innovation program 2014–2018 under grant agreement no. 677774 (European Research Council [ERC] Starting Grant: TEMPO) to R.B.J.B.

Acknowledgements. For access to specimens, we thank Judith White and Jo Cooper (Natural History Museum bird collection, Tring, UK), Janet Hinshaw (University of Michigan Museum of Zoology, Ann Arbour, Michigan), Mathew Lowe and Mike Brooke (University Museum of Zoology, Cambridge, UK), Mark Carnall and Eileen Westwig (Oxford University Museum of Natural History, Oxford, UK), Kristof Zyskowski (Yale Peabody Museum, New Haven, Connecticut), Ben Marks and John Bates (Field Museum of Natural History, Chicago). For access to CT scanning facilities we thank Ketura Smithson (Cambridge Biotomography Centre), Tom Davies, Ben Moon and Liz Martin-Silverstone (University of Bristol), Vincent Fernandez (Natural History Museum), April Neander and Zhe-Xi Luo (University of Chicago PaleoCT), and Matt Friedman (University of Michigan).

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
