## [Peer Review File · Proceedings of the Royal Society B: Biological Sciences]

Review History

RSPB-2020-1541.R0 (Original submission)

Review form: Reviewer 1

Recommendation

Major revision is needed (please make suggestions in comments)

Scientific importance: Is the manuscript an original and important contribution to its field?

Good

General interest: Is the paper of sufficient general interest?

Good

Quality of the paper: Is the overall quality of the paper suitable?

Good

Is the length of the paper justified?

Yes

Should the paper be seen by a specialist statistical reviewer?

No

Do you have any concerns about statistical analyses in this paper? If so, please specify them explicitly in your report.

No

It is a condition of publication that authors make their supporting data, code and materials available - either as supplementary material or hosted in an external repository. Please rate, if applicable, the supporting data on the following criteria.

Is it accessible?

Yes

Is it clear?

Yes

Is it adequate?

Yes

Do you have any ethical concerns with this paper?

No

Comments to the Author

The authors presented a deep and detailed analysis of the evolutionary morphology and modularity of the avian neck. By using geometric morphometrics and phylogenetic comparative methods, their approach allows to quantify inter- as well as intra-regional differences across a large set of bird species. My major concern is that the authors are somewhat over-enthusiastic about the findings and their inferred impact, particularly as this study and the results are in several aspects quite similar to the recent studies of Bohmer and colleagues (isometric scaling, minor impact of ecology, “deviations” in some specialists, etc.; Bohmer et al. 2019 R Soc Open Sci; Terray et al. 2020 Evol Biol – although different approaches were used in these studies). This is expressed by the extensive use of phrases like “unparalleled levels” or “we show, for the first time” across the manuscript. I agree, the findings add new insights in the evolvability of the avian cervical spine but these are mostly details (which are nevertheless highly interesting for researcher focused on the evolution of the vertebrate axial skeleton). For readers outside the field, however, the current manuscript somewhat implies that the avian neck has rarely been studied before the presented work (see details regarding introduction). However, research on birds’ neck has a long(er) tradition starting in in the first half of the 20th century (Boas), a peak in the 1990s and early 2000s (e.g. Bout, Zweers, van der Leeuw) and a more recent revival (e.g. Bohmer, Kambic and colleagues). This is not surprising as the avian neck is attractive for biologist from different aspects: the variability in cervical number allows to study the evolutionary and biomechanical consequences of “adding/removing” vertebrae, the complex kinematics provide insights into motor controls of long kinematic chains (and its potential translation into bionics/robotics) and avian necks are informative for understanding head posture and movement in dinosaur fossils. The authors indeed accounts for the huge amount of previous research when looking into the references (although few important sources are missing, see details below) but missed the fact that different approaches to the avian neck likely lead to different results. For this reason, I recommend to discuss why the author’s results are different from previous studies (not only stating that they are and previous studies missed to notice this or that) and more clearly show which are not.

The Introduction needs some major improvements. Most importantly, the introduction misses to summarize previous work on the modularity of the avian neck. Basic modularity has already been proposed and quantified by Boas in 1929 (Biologisch-anatomische Studien über den Hals der Vögel – although in German it is frequently cited and summarized in studies on the neck of birds and mammals). The functional morphology of the modularized avian neck was later assessed in different lineages (e.g. in owls and penguins; Krings et al. 2014 Plos One, Guinard & Marchand 2010 Evol Biol). In addition, avian neck modularity has recently extensively been studied using geometric morphometrics by Terray et al. 2020. This state of knowledge has to be presented in order to get what really are new findings of the presented study – even if that means

it might not look that innovative and impactful anymore.

P2L36: the first sentence tries to highlight the importance of the unresolved question but feels quite unrelated to the following ones. I recommend removing it and directly start with the avian neck.

P2L43ff: Terray et al 2020 specifically compared and discussed feeding ecology and its impact on neck modularity in birds

P2/3L45ff: indeed, the mammalian cervical spine recently got more attention from a morpho-functional and evolutionary view (Randau, Arnold etc.) but before the focus was completely on the developmental basis of the constraint to seven cervical vertebrae. As shown above, morpho-functional studies of birds have a much longer history. What is completely missing is the literature on dinosaur necks (particularly sauropods) which have extensively been studied by paleontologists and morphologists to reveal their head/neck posture. The same is true for the neck of extant and fossil turtles, in which it is even of systematic value (cryptodiran vs pleurodiran turtles) -see extensive studies of Werneburg and colleagues for neck modularity/regionalization, its evolution and its relationship to skull morphology. Altogether, this paragraph needs intensive revision.

P3L55ff: many tetrapod lineages show adaptation of their neck to ecology. Think about cervical fusion in fossorial or aquatic species (for a good review see VanBuren & Evans 2017 Biol Rev), shortening/flattening of vertebral centra in saltatorial aquatic mammals, shape and size of spinous processes in large headed/horned/antlered mammals, ... It's again a case of over-highlighting the study object.

P4L75: Why does the last sentence refer to reference 27? Basically, the reference shows that ecology/behavior-related morphological variation has previously been shown in mammals and is therefore not surprising in birds (thus it contradicts the sentence in which it is cited).

The Method section is good and needs minor improvements only:

P4L87: it is not completely clear why a suite of qualitative characters was recorded in addition to the Procrustes coordinates. What do these characters capture in detail that is not captured by landmarks? I first thought it is something like actual vertebral body length but as these characters are collected after the Procrustes analysis this information is lost. Please add more details.

P5L116: it is written that the data were subjected to a Procrustes analysis for the PTA analysis. Does this mean there was a second Procrustes fit in addition to the one mentioned before? Please clarify.

P6L120f: if I understand right the PTA is based on connecting the mean of one region across ecologically similar birds with the mean of the next region. But each bird's regional shape already is the mean of the vertebrae within this region - basically resulting in the comparison of means of means. Is this really reasonable. Are these means of means still biologically relevant. Wouldn't it be better to always take the middle vertebrae of each region and then use the mean of these representative vertebrae for region# across ecologically similar species?

P7f149: scaling and ecology are labeled as extrinsic factors in this sentence. In the rest of the manuscript, however, size is used as an intrinsic factor. Please clarify.

The Result section is quite long and as a reader one gets a little bit lost in details. The impact of extrinsic and intrinsic factors might be better presented in a table to show which factor best explains variability in which module. Morphological details and statistics could then be limited to the supplements and only most major trends could be summarized in the main text. As the journal usually publishes shorter manuscripts addressed to a wide readership, this could help to

reduce length and to increase readability for researcher outside the field.

As with the Results, the Discussion section loses itself a little bit in details. As the introduction missed to introduce details on previous finding on neck modularity in birds and mammals, it is hard for the reader to really asses what is a new finding and what is the impact of certain findings. The discussion needs a clear statement on which results agree with previous findings, which are contrary and which are new. In the next step, it would be useful to summarize in which features the avian neck is different to the mammalian neck (despite the variability in vertebral number). It is not sufficient just to state that birds are different. And in several instances, your findings even reveal similarities to mammals: neck elongation is primarily achieved by increasing midcervical region rather than upper or lower ones, mechanically demanding ecological behavior is associated with deviation from a more conserved ground plan etc. (Arnold et al. 2017 Evolution).

Moreover, some sections of the discussion are very speculative. Conservation in inter-regional morphology (P15L341ff) is speculated to be based on the optimization of the neck as a “hand” (i.e. optimizing motor pattern and control). As (almost) all mammals have only seven cervical vertebrae, this has been proposed as an explanation for conserved modular differences in vertebral morphology. However, such a functional explanation is hardly applicable to birds with varying number of cervical vertebrae. A conserved pattern of Hox gene expression might also be a good explanation.

Care should also be taken when explaining modifications related to carnivory. From a biomechanical point of view one would of course expect modifications in the lower cervical spine as even small changes in inter-vertebral mobility in this region would result in large changes in angular excursion of the head. However, carnivory does not always equal carnivory – even among vultures there are big differences on how they obtain food from carrion which are reflected their neck morphology (Bohmer et al. 2020 J Anat). Therefore, the authors should explicitly refer to the limitations of such inferences.

Review form: Reviewer 2

Recommendation

Accept with minor revision (please list in comments)

Scientific importance: Is the manuscript an original and important contribution to its field?

Excellent

General interest: Is the paper of sufficient general interest?

Acceptable

Quality of the paper: Is the overall quality of the paper suitable?

Good

Is the length of the paper justified?

Yes

Should the paper be seen by a specialist statistical reviewer?

Yes

Do you have any concerns about statistical analyses in this paper? If so, please specify them explicitly in your report.

No

It is a condition of publication that authors make their supporting data, code and materials available - either as supplementary material or hosted in an external repository. Please rate, if applicable, the supporting data on the following criteria.

Is it accessible?

Yes

Is it clear?

Yes

Is it adequate?

Yes

Do you have any ethical concerns with this paper?

No

Comments to the Author

The paper investigates the relationship between avian neck morphology and ecology such as feeding and locomotion. This is an area that is seeing interest given recent developments in detecting regionalization and increased interest in the axial skeleton. In general I believe the paper is strong and covers patterns which are not in the current literature.

My largest issue with the paper as it stands is that in my opinion the discussion of neck length and head/body mass is too strongly simplistic. There is definitely a relationship between neck length and mass (Fig 3), but body/head mass are poor predictors of neck length, particularly when it is noted that the plots are log transformed. The confidence intervals for head mass look to maybe capture ~30% of the data. I expect biological data to be messy with low explanatory power, but since this is a comparison entirely within (an admittedly diverse) single clade, some discussion of other possible factors I think is reasonable. I agree that the data the authors gathered do not explain deviation from this relationship well, so what other factors might be contributing?

Following the Handbook of Avian Anatomy I recommend using cranial/caudal instead of anterior/posterior to eliminate possible confusion.

100: Is this supposed to be 48? Or is the earlier number of 48 species incorrect? Or were masses not collected for every species?

624: Missing reference information.

Figure 2: Trajectories are hard to see on these small plots.

Is Figure 3 referenced in the text?

Decision letter (RSPB-2020-1541.R0)

03-Aug-2020

Dear Dr Marek:

I am writing to inform you that we have now obtained responses from referees on manuscript RSPB-2020-1541 entitled "Evolutionary versatility of the avian neck" which you submitted to Proceedings B.

Unfortunately, on the advice of the Associate Editor and the referees, your manuscript has been rejected following full peer review. Competition for space in Proceedings B is currently extremely severe, as many more manuscripts are submitted to us than we have space to print. We are therefore only able to publish those that are exceptional, convincing and present significant advances of broad interest, and must reject many good manuscripts.

On a more positive note, based on the advice we have received, we would like to offer you the opportunity to transfer your manuscript file to another Royal Society journal, Royal Society Open Science. Royal Society Open Science is a fast, open journal publishing high-quality research across all of science and mathematics. The journal operates objective peer review, optional open peer review, and will publish any article deemed to sufficiently advance the field by the reviewers and editors, leaving judgement of potential impact of the work to the reader. The journal publishes Registered Reports and encourages the submission of negative results. You can find out more about the scope of the journal and the benefits of publication here <https://royalsocietypublishing.org/journal/rsos>

If you wish to have your manuscript transferred to Royal Society Open Science please ensure that you revise your text to address all of the reviewers' comments relating to scientific soundness. Please particularly ensure that your conclusions do not overstate the results of your study. Once submitted to Royal Society Open Science your manuscript will be assessed by an Associate Editor who will decide whether further reviewer advice is required. If no further advice is needed and all of your revisions are satisfactory your manuscript will be immediately accepted for publication.

If you agree to transfer your paper, and it is accepted for publication, you will be asked to pay the article processing charge, unless you request a waiver and this is approved by Royal Society Publishing. You can find out more about the charges at <https://royalsocietypublishing.org/rsos/charges>.

You can approve or reject this transfer using the links below:

Approve transfer - *** PLEASE NOTE: This is a two-step process. After clicking on the link, you will be directed to a webpage to confirm. ***

https://mc.manuscriptcentral.com/prsb?URL_MASK=ebdc1dab1e3d43f79b0027ff27907cd8
After approving the transfer you will need to log in to your Royal Society Open Science author centre (<https://mc.manuscriptcentral.com/rsos>) to complete your the submission. At this stage you will have chance to address any of the reviewers' or editor's concerns.

Reject transfer - *** PLEASE NOTE: This is a two-step process. After clicking on the link, you will be directed to a webpage to confirm. ***

https://mc.manuscriptcentral.com/prsb?URL_MASK=135ca8b7d8204ba581c4a191581ce9d3

or by clicking 'approve' or 'reject' in your Author Center.

Once you have approved the transfer you will be prompted to complete the transfer of your article via the Royal Society Open Science submission system.

Please find below the comments received from referees concerning your manuscript, not including confidential reports to the Editor. If you approve transfer to Royal Society Open Science, these reviews will accompany your paper.

Thank you for your interest in Proceedings B.

Sincerely,
Proceedings B
mailto: proceedingsb@royalsociety.org

Associate Editor
Comments to Author:

The reviewers agree that the study is well conducted, methodologically strong, and scientifically important. They expressed some concern that the existing literature and scientific context are not thoroughly considered, and the introduction and discussion should be revised to address these deficits. This includes interpretation of the relationship between neck length and mass (as noted by Reviewer 2). The results also might be presented more clearly and succinctly, with some details moved to SI.

Reviewer(s)' Comments to Author:

Referee: 1

Comments to the Author(s)

The authors presented a deep and detailed analysis of the evolutionary morphology and modularity of the avian neck. By using geometric morphometrics and phylogenetic comparative methods, their approach allows to quantify inter- as well as intra-regional differences across a large set of bird species. My major concern is that the authors are somewhat over-enthusiastic about the findings and their inferred impact, particularly as this study and the results are in several aspects quite similar to the recent studies of Bohmer and colleagues (isometric scaling, minor impact of ecology, “deviations” in some specialists, etc.; Bohmer et al. 2019 R Soc Open Sci; Terray et al. 2020 Evol Biol – although different approaches were used in these studies). This is expressed by the extensive use of phrases like “unparalleled levels” or “we show, for the first time” across the manuscript. I agree, the findings add new insights in the evolvability of the avian cervical spine but these are mostly details (which are nevertheless highly interesting for researcher focused on the evolution of the vertebrate axial skeleton). For readers outside the field, however, the current manuscript somewhat implies that the avian neck has rarely been studied before the presented work (see details regarding introduction). However, research on birds’ neck has a long(er) tradition starting in in the first half of the 20th century (Boas), a peak in the 1990s and early 2000s (e.g. Bout, Zweers, van der Leeuw) and a more recent revival (e.g. Bohmer, Kambic and colleagues). This is not surprising as the avian neck is attractive for biologist from different aspects: the variability in cervical number allows to study the evolutionary and biomechanical consequences of “adding/removing” vertebrae, the complex kinematics provide insights into motor controls of long kinematic chains (and its potential translation into bionics/robotics) and avian necks are informative for understanding head posture and movement in dinosaur fossils. The authors indeed accounts for the huge amount of previous research when looking into the references (although few important sources are missing, see details below) but missed the fact that different approaches to the avian neck likely lead to different results. For this reason, I recommend to discuss why the author’s results are different from previous studies (not only stating that they are and previous studies missed to notice this or that) and more clearly show which are not.

The Introduction needs some major improvements. Most importantly, the introduction misses to summarize previous work on the modularity of the avian neck. Basic modularity has already been proposed and quantified by Boas in 1929 (*Biologisch-anatomische Studien über den Hals der Vögel* – although in German it is frequently cited and summarized in studies on the neck of birds and mammals). The functional morphology of the modularized avian neck was later assessed in different lineages (e.g. in owls and penguins; Krings et al. 2014 Plos One, Guinard & Marchand 2010 Evol Biol). In addition, avian neck modularity has recently extensively been studied using geometric morphometrics by Terray et al. 2020. This state of knowledge has to be presented in order to get what really are new findings of the presented study – even if that means it might not look that innovative and impactful anymore.

P2L36: the first sentence tries to highlight the importance of the unresolved question but feels quite unrelated to the following ones. I recommend removing it and directly start with the avian neck.

P2L43ff: Terray et al 2020 specifically compared and discussed feeding ecology and its impact on neck modularity in birds

P2/3L45ff: indeed, the mammalian cervical spine recently got more attention from a morpho-functional and evolutionary view (Randau, Arnold etc.) but before the focus was completely on the developmental basis of the constraint to seven cervical vertebrae. As shown above, morpho-functional studies of birds have a much longer history. What is completely missing is the literature on dinosaur necks (particularly sauropods) which have extensively been studied by paleontologists and morphologists to reveal their head/neck posture. The same is true for the neck of extant and fossil turtles, in which it is even of systematic value (cryptodiran vs pleurodiran turtles) -see extensive studies of Werneburg and colleagues for neck modularity/regionalization, its evolution and its relationship to skull morphology. Altogether, this paragraph needs intensive revision.

P3L55ff: many tetrapod lineages show adaptation of their neck to ecology. Think about cervical fusion in fossorial or aquatic species (for a good review see VanBuren & Evans 2017 Biol Rev), shortening/flattening of vertebral centra in saltatorial aquatic mammals, shape and size of spinous processes in large headed/horned/antlered mammals, ... It's again a case of over-highlighting the study object.

P4L75: Why does the last sentence refer to reference 27? Basically, the reference shows that ecology/behavior-related morphological variation has previously been shown in mammals and is therefore not surprising in birds (thus it contradicts the sentence in which it is cited).

The Method section is good and needs minor improvements only:

P4L87: it is not completely clear why a suite of qualitative characters was recorded in addition to the Procrustes coordinates. What do these characters capture in detail that is not captured by landmarks? I first thought it is something like actual vertebral body length but as these characters are collected after the Procrustes analysis this information is lost. Please add more details.

P5L116: it is written that the data were subjected to a Procrustes analysis for the PTA analysis. Does this mean there was a second Procrustes fit in addition to the one mentioned before? Please clarify.

P6L120f: if I understand right the PTA is based on connecting the mean of one region across ecologically similar birds with the mean of the next region. But each bird's regional shape already is the mean of the vertebrae within this region - basically resulting in the comparison of means of means. Is this really reasonable. Are these means of means still biologically relevant. Wouldn't it be better to always take the middle vertebrae of each region and then use the mean of these representative vertebrae for region# across ecologically similar species?

P7f149: scaling and ecology are labeled as extrinsic factors in this sentence. In the rest of the manuscript, however, size is used as an intrinsic factor. Please clarify.

The Result section is quite long and as a reader one gets a little bit lost in details. The impact of extrinsic and intrinsic factors might be better presented in a table to show which factor best explains variability in which module. Morphological details and statistics could then be limited to the supplements and only most major trends could be summarized in the main text. As the

journal usually publishes shorter manuscripts addressed to a wide readership, this could help to reduce length and to increase readability for researcher outside the field.

As with the Results, the Discussion section loses itself a little bit in details. As the introduction missed to introduce details on previous finding on neck modularity in birds and mammals, it is hard for the reader to really asses what is a new finding and what is the impact of certain findings. The discussion needs a clear statement on which results agree with previous findings, which are contrary and which are new. In the next step, it would be useful to summarize in which features the avian neck is different to the mammalian neck (despite the variability in vertebral number). It is not sufficient just to state that birds are different. And in several instances, your findings even reveal similarities to mammals: neck elongation is primarily achieved by increasing midcervical region rather than upper or lower ones, mechanically demanding ecological behavior is associated with deviation from a more conserved ground plan etc. (Arnold et al. 2017 Evolution).

Moreover, some sections of the discussion are very speculative. Conservation in inter-regional morphology (P15L341ff) is speculated to be based on the optimization of the neck as a “hand” (i.e. optimizing motor pattern and control). As (almost) all mammals have only seven cervical vertebrae, this has been proposed as an explanation for conserved modular differences in vertebral morphology. However, such a functional explanation is hardly applicable to birds with varying number of cervical vertebrae. A conserved pattern of Hox gene expression might also be a good explanation.

Care should also be taken when explaining modifications related to carnivory. From a biomechanical point of view one would of course expect modifications in the lower cervical spine as even small changes in inter-vertebral mobility in this region would result in large changes in angular excursion of the head. However, carnivory does not always equal carnivory – even among vultures there are big differences on how they obtain food from carrion which are reflected their neck morphology (Bohmer et al. 2020 J Anat). Therefore, the authors should explicitly refer to the limitations of such inferences.

Referee: 2

Comments to the Author(s)

The paper investigates the relationship between avian neck morphology and ecology such as feeding and locomotion. This is an area that is seeing interest given recent developments in detecting regionalization and increased interest in the axial skeleton. In general I believe the paper is strong and covers patterns which are not in the current literature.

My largest issue with the paper as it stands is that in my opinion the discussion of neck length and head/body mass is too strongly simplistic. There is definitely a relationship between neck length and mass (Fig 3), but body/head mass are poor predictors of neck length, particularly when it is noted that the plots are log transformed. The confidence intervals for head mass look to maybe capture ~30% of the data. I expect biological data to be messy with low explanatory power, but since this is a comparison entirely within (an admittedly diverse) single clade, some discussion of other possible factors I think is reasonable. I agree that the data the authors gathered do not explain deviation from this relationship well, so what other factors might be contributing?

Following the Handbook of Avian Anatomy I recommend using cranial/caudal instead of anterior/posterior to eliminate possible confusion.

100: Is this supposed to be 48? Or is the earlier number of 48 species incorrect? Or were masses not collected for every species?

624: Missing reference information.

Figure 2: Trajectories are hard to see on these small plots.

Is Figure 3 referenced in the text?

Author's Response to Decision Letter for (RSPB-2020-1541.R0)

See Appendix A.

RSPB-2020-3150.R0

Review form: Reviewer 1

Recommendation

Accept with minor revision (please list in comments)

Scientific importance: Is the manuscript an original and important contribution to its field?

Excellent

General interest: Is the paper of sufficient general interest?

Good

Quality of the paper: Is the overall quality of the paper suitable?

Excellent

Is the length of the paper justified?

Yes

Should the paper be seen by a specialist statistical reviewer?

No

Do you have any concerns about statistical analyses in this paper? If so, please specify them explicitly in your report.

No

It is a condition of publication that authors make their supporting data, code and materials available - either as supplementary material or hosted in an external repository. Please rate, if applicable, the supporting data on the following criteria.

Is it accessible?

No

Is it clear?

Yes

Is it adequate?

Yes

Do you have any ethical concerns with this paper?

No

Comments to the Author

I am glad to see that the authors presented a thoroughly revised manuscript. Actually, I agree with most of the authors' arguments but they seemed to have interpreted my previous recommendations more critically/negatively than they were intended. I did not intend to downgrade the results of the study (it's the details in which most researchers are right interested in), I just wanted the authors to explain them in way more "graspable" for readers outside the field.

I want to especially highlight the revised introduction which now leads readers outside the field of axial evolution much better to the research question. Details I have questioned before in the method section are also much clearer now. As recommended, phrases like "for the first time" have been reduced in number and, most importantly, the conceptual differences to previous work of Terray and Bohmer are now explicitly outlined (I want to underline that I am not a member of these authors or their research group). The rearrangement of the result and discussion section makes them much more readable and provide good take home messages for a broader readership.

Overall, the authors really used the second chance given by the editor and present now a manuscript with high quality for which I have now further complains except one question:

Will the surface models and landmark data files be publicly available? This would reduce the need for CT scanning museum specimens in future studies by other researchers. It further enables reproducing or expanding some of the analyses (e.g., PTA). I therefore highly recommend this.

Decision letter (RSPB-2020-3150.R0)

29-Jan-2021

Dear Dr Marek

I am pleased to inform you that your manuscript RSPB-2020-3150 entitled "Evolutionary versatility of the avian neck" has been accepted for publication in Proceedings B.

The referee(s) have recommended publication, but also suggest some minor revisions to your manuscript. Therefore, I invite you to respond to the referee(s)' comments and revise your manuscript. Because the schedule for publication is very tight, it is a condition of publication that you submit the revised version of your manuscript within 7 days. If you do not think you will be able to meet this date please let us know.

Sincerely,

Dr Sasha Dall
mailto: proceedingsb@royalsociety.org

Associate Editor
Board Member
Comments to Author:

The reviewer is satisfied with the new version of the manuscript and feels that the broader significance of the study is now very clear. However, they bring up an important point: in accordance with our policies, underlying data (in this case surface/landmark data) need to be made accessible at the time of publication unless there is a compelling reason not to do so.

Reviewer(s)' Comments to Author:

Referee: 1

Comments to the Author(s).

I am glad to see that the authors presented a thoroughly revised manuscript. Actually, I agree with most of the authors' arguments but they seemed to have interpreted my previous recommendations more critically/negatively than they were intended. I did not intend to downgrade the results of the study (it's the details in which most researchers are right interested in), I just wanted the authors to explain them in a way more "graspable" for readers outside the field.

I want to especially highlight the revised introduction which now leads readers outside the field of axial evolution much better to the research question. Details I have questioned before in the method section are also much clearer now. As recommended, phrases like "for the first time" have been reduced in number and, most importantly, the conceptual differences to previous work of Terray and Bohmer are now explicitly outlined (I want to underline that I am not a member of these authors or their research group). The rearrangement of the result and discussion section makes them much more readable and provides good take-home messages for a broader readership.

Overall, the authors really used the second chance given by the editor and present now a manuscript with high quality for which I have no further complaints except one question:

Will the surface models and landmark data files be publicly available? This would reduce the need for CT scanning museum specimens in future studies by other researchers. It further enables reproducing or expanding some of the analyses (e.g., PTA). I therefore highly recommend this.

Author's Response to Decision Letter for (RSPB-2020-3150.R0)

See Appendix B.

Decision letter (RSPB-2020-3150.R1)

03-Feb-2021

Dear Dr Marek

I am pleased to inform you that your manuscript entitled "Evolutionary versatility of the avian neck" has been accepted for publication in Proceedings B.

Open Access

Paper charges

Sincerely,

Appendix A

Response to Reviewer and Editor Comments

We are very grateful to the editor and reviewers for taking time to read our manuscript, and for the critical but thoughtful comments, which will greatly improve the paper. Please find our response to these comments below. Reviewer comments are in blue. Our response in black, with quotes in green.

Associate Editor

Comments to Author:

The reviewers agree that the study is well conducted, methodologically strong, and scientifically important. They expressed some concern that the existing literature and scientific context are not thoroughly considered, and the introduction and discussion should be revised to address these deficits. This includes interpretation of the relationship between neck length and mass (as noted by Reviewer 2). The results also might be presented more clearly and succinctly, with some details moved to SI.

In summary, we have carefully edited the introduction and discussion section, and shifted aspects of our results to the supplementary information. We have expanded our discussion and interpretation of the relationship between neck length and head/body mass and the scatter of the data following the comment of reviewer 2. We have provided more detailed explanations of these changes below next to the specific comments by the reviewers.

Reviewer(s)' Comments to Author:

Referee: 1

Comments to the Author(s)

Summary of response to reviewer 1

We thank the reviewer for clearly considering our study and its place in the literature. We agree whole-heartedly that study of the avian neck has a long history, which reflects the fact (as the reviewer points out) that it is a unique and fascinating system that is of interest to range of biological scientists. The vast majority of Referee 1's comments relate to the same point, specifically to how our analyses compare to previous work. The comparisons made by the reviewer often carry the implication that our study is very similar (perhaps even overlapping) with previous work on the avian neck, in particular two recent studies by the same research group (Bohmer et al. 2019; Terray et al. 2020) and that we have failed to acknowledge that fact in our manuscript. While we have made changes in our resubmission in response to these comments, we largely disagree with this point generally, and with many of the specific comments made in the review. We have replied to individual comments below, but for brevity and clarity we also summarise our response here in two main points:

(1) We cited 96 studies in our original submission. Where we feel it appropriate, we have expanded citations and the text to add a review-style element when referencing previous work. However, stylistically we feel that original research articles in Proceedings B should be driven by the aims and results of the study being presented and not by the need to review all past work in the general research area in great detail. The length of our original submission was only just below the 10-page limit for the journal and thus there was limited scope to add a large review element given the volume of data we present.

(2) We feel that while there is conceptual overlap between our work and some past studies, we present analyses (and deliver results) that have not been attempted in any previous work on the avian neck, and indeed perhaps the vertebrate neck more widely (at least in one single study/data set). These new analyses provide important novel insight into the construction of the avian neck and

how that construction has been shaped by interacting extrinsic (diet, locomotion) and intrinsic (body size, head size) factors. We have attempted to explain these new findings in the manuscript and where relevant in our responses below.

The authors presented a deep and detailed analysis of the evolutionary morphology and modularity of the avian neck. By using geometric morphometrics and phylogenetic comparative methods, their approach allows to quantify inter- as well as intra-regional differences across a large set of bird species. My major concern is that the authors are somewhat over-enthusiastic about the findings and their inferred impact, particularly as this study and the results are in several aspects quite similar to the recent studies of Bohmer and colleagues (isometric scaling, minor impact of ecology, “deviations” in some specialists, etc.; Bohmer et al. 2019 R Soc Open Sci; Terray et al. 2020 Evol Biol – although different approaches were used in these studies).

We agree that there are conceptual similarities, and in some specific cases overlap, between our work and those of Bohmer et al (2019) and Terray et al. (2020). However, there are also major fundamental differences in the data assembled, the statistical analyses carried out and (as a result of this) and the questions addressed and answered by these studies and ours. We explain these differences separately for the two studies, as follows -

Bohmer et al (2019): Contrary to the implication of this comment of the reviewer, in our first submission we explicitly acknowledged that our finding of isometry in neck length vs body size is consistent with the finding in Bohmer et al. (2019) in our discussion section:

“Avian neck length scales isometrically with head mass and body mass (Fig S2, Table S4). This differs from other groups of vertebrates (e.g. negative allometry in mammals (18)), as was also reported in other recent work (5).”

Where reference 5 was Bohmer et al. (2019). At no point in our paper do we put this result forward as our landmark finding, and at no point is it associated with the potentially hyperbolic phrases (“*for the first time*”) Reviewer 1 goes on to mention subsequently in their review. It was necessary that we pointed to this finding in our abstract because we also find that the neck scales isometrically with head size (which is a novel aspect of our study). However, it is a very minor part of our work overall and criticism of its novelty does not detract from the significant novel aspects of our study overall. Nevertheless, our scaling section does deliver new and significant findings that the reviewer has not mentioned. First, we also present an analysis of how head mass with body mass and relative to the neck, which Bohmer et al. (2019) do not. We would argue that the interaction between head mass and neck length are equally interesting and important to study because of the strong selective pressure/inter-action between the mass of the head and length of the neck in vertebrates, particularly in animals that lack grasping forelimbs and are more reliant on the head and neck for environmental interaction. We also statistically examine how neck elongation is achieved in birds, which Bohmer et al. (2019) do not. These points collectively represent an advance in understanding of the drivers of avian length neck evolution and the occurrence of integrated evolution among body regions.

Terray et al. (2020): The reviewer is correct in so far as Terray et al. (2020) do examine modularity in the avian neck using geometric morphometrics. We already acknowledged this explicitly in the second sentence of our introduction in the first submission where we stated “*The avian neck is a highly modular structure (3)...*” where reference 3 was Terray et al. (2020). At no point in our introduction or discussion do we argue that the novelty of our study comes from taking a modular approach and/or using geometric morphometrics specifically. However, we are the first to show that a large taxonomically and ecologically diverse sample of birds have 5 morphological regions whose boundaries are consistent with *Hox* gene expression limits, which Terray et al. (2020) do not attempt to do (they use their own morphological system, which has not been linked to genetics). We

then use this homology-based framework as a basis to conduct phylogenetically informed statistical correlations with intrinsic (neck length, body size, head, region size) and extrinsic (locomotion, diet/feeding ecology) factors that might be expected to have exerted selective pressure on neck morphology at different hierarchical scales (whole neck down to intra-regional), which Terray et al. (2020) do not. Our data set is more than 3 times larger than that of Terray et al. (2020). We have covered all major taxonomic sub-groups, locomotor and tropic ecologies occupied by birds in our data set. Fundamentally this allows us to address the aforementioned questions about morphological diversity in the avian neck that previously studies have not attempted to do. Terray et al. (2020) is an excellent study, but we do not feel it detracts from the novelty and importance of our study for these reasons. The goal and results delivered by Terray et al (2020) are summarised in their own introduction section as follows: “*The aim of this study was to reveal how shape diversity is structured in bird necks by studying the patterns of modularity at the interspecific level....For each module, the phylogenetic signal was assessed, and postural properties in a relaxed posture were studied.*” So, in their actual analysis, Terray et al (2020) derived their own sub-regions for avian neck in a small sample of birds (adding to the many different regional schemes previously published, as the reviewer points out), they test how phylogeny is influencing their particular signal in terms of sub-regions, and they present in-depth analysis of the static “*osteological neutral posture*” of vertebrae in their defined regions (this analysis of “posture” actually represents the bulk of their paper and is not something we are at all interested in here). Terray et al. (2020) do not do any statistics related to either locomotor or dietary ecology, or intrinsic factors (body size, neck length, region size etc.). Therefore, they do not even attempt the kind of phylogenetically informed, formal statistical analysis of intrinsic (size) and extrinsic (ecological) adaptations in birds necks that we do in our study. They therefore offer no results that overlap with our most significant findings. We emphasise these differences further below in response to another comment about Terray et al. (2020) by the reviewer.

This is expressed by the extensive use of phrases like “unparalleled levels” or “we show, for the first time” across the manuscript. I agree, the findings add new insights in the evolvability of the avian cervical spine but these are mostly details (which are nevertheless highly interesting for researcher focused on the evolution of the vertebrate axial skeleton). For readers outside the field, however, the current manuscript somewhat implies that the avian neck has rarely been studied before the presented work (see details regarding introduction). However, research on birds’ neck has a long(er) tradition starting in in the first half of the 20th century (Boas), a peak in the 1990s and early 2000s (e.g. Bout, Zweers, van der Leeuw) and a more recent revival (e.g. Bohmer, Kambic and colleagues).

In hindsight, we acknowledge there was perhaps a little too much hyperbole (e.g. an unnecessary number of “*for the first time*”) in certain sections of our manuscript. We have removed most of these phrases in our resubmission and generally sought to clarify firm vs. cautious interpretations throughout. However, we do not believe that we have misrepresented or omitted previous work to make our results appear more novel, and we emphasise again that the novel aspects we highlight are not “*mostly details.*” This is clearly demonstrated by the fundamental differences we highlight above (and below) between our study and that of Terray et al. (2020), which the reviewer clearly feels is most similar to our present work. The studies cited here (Bout, Zweers, van der Leeuw; Boas; Bohmer, Kambic and colleagues) are all excellent studies, but they are either entirely qualitative/descriptive pieces of comparative anatomy, or they are studies of a single taxon dealing with regional variation in the motion of the neck, rather than its morphology and (statistical tests of) the relationship of that morphology to multiple intrinsic and extrinsic biological/ecological factors as we present here. Many of these previous works have framed hypotheses, but have lacked statistical hypotheses tests based on rigorous comparative data at many scales of observation. Uniquely, we quantitatively and statistically examine morphological diversity and its interaction with phylogeny, body size, locomotor and dietary ecology in a large data set of birds (an excellent

“case study” group, as the reviewer notes below), and thus our work stands out in the avian literature and arguably vertebrate neck literature more widely. We deliver important and fundamental findings (through quantitative analyses) about factors that have shaped the morphological organisation of the avian neck at multiple scales that previous studies have not attempted.

This is not surprising as the avian neck is attractive for biologist from different aspects: the variability in cervical number allows to study the evolutionary and biomechanical consequences of “adding/removing” vertebrae, the complex kinematics provide insights into motor controls of long kinematic chains (and its potential translation into bionics/robotics) and avian necks are informative for understanding head posture and movement in dinosaur fossils. The authors indeed accounts for the huge amount of previous research when looking into the references (although few important sources are missing, see details below) but missed the fact that different approaches to the avian neck likely lead to different results. For this reason, I recommend to discuss why the author’s results are different from previous studies (not only stating that they are and previous studies missed to notice this or that) and more clearly show which are not.

We thank the reviewer for the acknowledgement that the avian neck is excellent case study system in terms of understanding what drives form-function diversity in the vertebrate neck and therefore will be of broad interest to a range of biologists. We accept the general point that, at certain points in the manuscript, it would be beneficial to elaborate a little more on the findings of previous studies to make the novelty of our findings clearer, but would reiterate again that we feel a long detail review is beyond the scope of this paper.

However, the reviewer has stated the following above: “*missed the fact that different approaches to the avian neck lead to different results*” and “*discuss why the author’s results are different from previous studies (not only stating that they are and previous studies missed to notice this or that).*” In response to these, we are not completely clear what is meant by the first comment. Our study characterises the morphology of the avian neck on multiple scales, using appropriate (phylogenetically-informed, multivariate) statistical approaches that ask specific questions about evolutionary linkages among body regions and between morphology and ecology. Our results differ from those of some previous studies because we leverage considerably more evidence. We emphasise throughout that our study is the first to quantitatively and statistically examine morphological diversity and its interaction with phylogeny, body size, locomotor and dietary ecology in bird necks, so for the most part, there is no comparison of “*different methods giving different results*” to be made. As we acknowledge above and discuss again in more detail in response to the next comment below, others have discussed regionalisation before, and Terray et al. (2020) used a different approach to ours to quantitatively define the number of regions in a smaller sample of birds. It will be difficult to discuss these in more detail without appearing to criticise these studies (e.g. for being purely qualitative, using small data sets or for using quantitative approaches that ignore fundamental genetic homology and developmental control of regions), which we would rather not do unnecessarily. This follows into the second claim by the reviewer above, which we also believe to be unfair: at no point do we criticise a previous study in our manuscript. At no point do we accuse a previous study of simply “*missing things*” in order to make our work look more important. We could have done this (e.g. by pointing out that many previous studies have been qualitative or focused on a single species, used small sample sizes etc.) but we preferred to concentrate on positive results that our analyses revealed rather than potential limitations in others. We have checked our first submission very carefully and, contrary to the reviewer’s comment, at no point do we go out of our way to criticise any particular prior study or body of work.

The Introduction needs some major improvements. Most importantly, the introduction misses to summarize previous work on the modularity of the avian neck. Basic modularity has already been

proposed and quantified by Boas in 1929 (Biologisch-anatomische Studien über den Hals der Vögel – although in German it is frequently cited and summarized in studies on the neck of birds and mammals). The functional morphology of the modularized avian neck was later assessed in different lineages (e.g. in owls and penguins; Krings et al. 2014 Plos One, Guinard & Marchand 2010 Evol Biol). In addition, avian neck modularity has recently extensively been studied using geometric morphometrics by Terray et al. 2020. This state of knowledge has to be presented in order to get what really are new findings of the presented study – even if that means it might not look that innovative and impactful anymore.

Our response to this comment is essentially the same as the previous comment: With the exception of Terray et al. (2020), all the papers cited here either qualitatively discuss regionalisation in bird necks (e.g. Boas) or represent one-taxon biomechanical studies of how the neck in particular species of birds moves (Krings et al. 2014, Guinard & Marchand 2010). We do not believe that detailed discussion of any of these studies would force us to acknowledge that our study is not “*innovative and impactful anymore.*” In our resubmission we have heavily edited the introduction in an attempt to reference these studies (and others) in a more elaborate way and distinguish them from our current study and what it attempts to do. We feel that acknowledging these studies more explicitly does not force us to alter any of the important sentences in the introduction that set up our novel study goals and findings, such as:

“However, no previous body of work has quantitatively investigated the ecomorphological signal in this variation, despite the clear functional significance and variability the avian neck displays.”

“In contrast, avian necks show great capacity for evolutionary variation, but the effects of ecomorphological and intrinsic constraints on avian neck evolution have not been quantified, and this represents a major gap in understanding of phenotypic plasticity of the vertebrate neck.”

“A key question concerns whether the phenotypic plasticity of the avian cervical column is driven by adaptive responses to extrinsic (ecological) factors, or by intrinsic (scaling) constraints.... This hypothesis has not previously been tested due the difficulty in comparing vertebral anatomy between species with different counts of cervical vertebrae, as the homology of individual vertebrae in this case is unclear.... Issues concerning homology between species with differing cervical counts could potentially be resolved by utilising regional morphology as a metric of study, however the hypothesis that five cervical regions are present across extant Aves has not been tested.”

P2L36: the first sentence tries to highlight the importance of the unresolved question but feels quite unrelated to the following ones. I recommend removing it and directly start with the avian neck.

For clarity, the sentence in question is this one:

“How, why and at what scale phenotypic plasticity arises in morphological structures are amongst the most important questions in evolutionary biology (1,2).”

While it would not be a major issue to remove this sentence, we are reluctant to do so because we feel this opening sentence provides some wider context to our specific work on the avian neck. We are ultimately attempting to understand “*how*” species of bird differ in neck morphology, “*why*” they differ (e.g. is it mainly ecology, body size, phylogenetic etc?) and “*at what scale*” the “*how*” and “*why*” are most strongly expressed (whole-neck, inter-regional, intra-regional etc.). We therefore feel this sentence is an appropriate starting point for the paper, although we have reworded it slightly in our resubmission.

P2L43ff: Terray et al 2020 specifically compared and discussed feeding ecology and its impact on

neck modularity in birds

Terray et al. (2020) do “*discuss*” feeding ecology and its relationship to modularity in birds, in both their introduction and discussion sections. But they do not present any test of the hypothesised relationship between the two (or any other factors, such as neck length, body mass etc. as we do). Indeed, in their discussion section Terray et al. (2020) very briefly (one paragraph) discuss the idea that the apparent lack of variation in their small sample of birds possibly means that feeding ecology has not strongly influenced neck morphology. However, it is just discussion, not analysis. They actually conclude this brief theoretical paragraph by stating that a new study (similar to ours) is required to understand the nature and magnitude of ecologically-driven adaptation in bird neck morphology:

“We can therefore hypothesize that ecological factors might apply heterogeneously along the vertebral column, according to phylogeny. To test this hypothesis, it would be interesting to expand our dataset with other species to obtain statistically testable ecological groups.”

Of all the studies the reviewer cites, Terray et al. (2020) is the most similar to ours, but Terray et al. (2020) themselves, in their own words, clearly emphasise that they have not attempted to deliver the quantitative analyses that we do deliver in our paper.

P2/3L45ff: indeed, the mammalian cervical spine recently got more attention from a morpho-functional and evolutionary view (Randau, Arnold etc.) but before the focus was completely on the developmental basis of the constraint to seven cervical vertebrae. As shown above, morpho-functional studies of birds have a much longer history. What is completely missing is the literature on dinosaur necks (particularly sauropods) which have extensively been studied by paleontologists and morphologists to reveal their head/neck posture. The same is true for the neck of extant and fossil turtles, in which it is even of systematic value (cryptodiran vs pleurodiran turtles) -see extensive studies of Werneburg and colleagues for neck modularity/regionalization, its evolution and its relationship to skull morphology. Altogether, this paragraph needs intensive revision.

The main goal of analysis is to characterise morphological variation in extant bird necks at the species level and to quantify its relationship to extrinsic and intrinsic biological factors and so setting up these goals was and is the main function of our introduction (and begins with the very first sentence of the paper, which makes it clear we’re focusing on phenotypic variation). As stated above, our paper is already very close to the 10-page limit for Proceedings B and so there is limited scope for adding detailed review elements. We initially excluded reference to the work on sauropod head and neck posture because it seemed too far removed from our specific analyses that deal with birds, their morphology and its relationship to ecology and body size (not to neck/head posture). It’s not that we have a vested interest in ignoring this work; indeed we could easily boost our own citation metrics by diverging into sauropod neck elongation (e.g. Bates et al. 2016, Royal Society Open Science), but we maintain the view that this work is not relevant enough to warrant a detailed review in our introduction. Similarly, we initially excluded the excellent work of Werneburg’s group on turtles because here we are focusing on phenotypic variation and quite specific ecological types, whereas the turtle work is focused at a much higher-level taxonomically and adaptively by attempting to understand the evolution of major neck retraction mechanisms in turtles (i.e. Cryptodira [side-necked] vs. Pleurodira [hidden-necked]). This body of work does not test for intrinsic and extrinsic (ecological) drivers of morphological variation in the same way we have in birds, although we do agree that by studying a major morpho-functional innovation they are examining adaptive responses in the neck in such a way that could be mentioned in our introduction. We therefore incorporated very brief mention of these works in our revised introduction section.

P3L55ff: many tetrapod lineages show adaptation of their neck to ecology. Think about cervical fusion in fossorial or aquatic species (for a good review see VanBuren & Evans 2017 Biol Rev), shortening/flattening of vertebral centra in saltatorial aquatic mammals, shape and size of spinous processes in large headed/horned/antlered mammals, ... It's again a case of over-highlighting the study object.

We agree that this sentence was poorly worded and has been removed from the revised introduction section. However, we were not trying to over-highlight our work: we didn't explicitly mention vertebral fusion and the review paper by Van Buren & Evans for the same reason mentioned above for Werneburg/turtles, i.e. it's not an example where species-level morphological variation has been quantified across a major group and links to intrinsic and extrinsic drivers statistically tested. We now cite this example alongside Werneburg/turtles where we highlight major "higher-level" analyses of singular evolutionary innovations in neck morphology.

P4L75: Why does the last sentence refer to reference 27? Basically, the reference shows that ecology/behavior-related morphological variation has previously been shown in mammals and is therefore not surprising in birds (thus it contradicts the sentence in which it is cited).

Reference 27 was cited to highlight an exemplar exception to the wider trend. The exact wording here was:

"...and one of the first demonstrations of ecologically 75 associated morphological variation in the tetrapod cervical column (27)."

With reference 27 cited to emphasise the "*one of the first*" but not the first aspect of the sentence. However, this sentence has been removed in our revised introduction and so this is no longer an issue.

The Method section is good and need minor improvements only:

Thank you. We are glad it is largely clear.

P4L87: it is not completely clear why a suite of qualitative characters was recorded in addition to the Procrustes coordinates. What do these characters capture in detail that is not captured by landmarks? I first thought it is something like actual vertebral body length but as these characters are collected after the Procrustes analysis this information is lost. Please add more details.

We used the same morphological scheme used by Böhmer et al (2015) on the chicken because this scheme (combining landmarks and qualitative characters) yielded regional distinctions that match *Hox* gene expression limits. These characters include the presence and absence of osteological features, such as a ventral keel, a bifurcated neural spine and muscle insertion points that vary within each cervical series and could not be captured by homologous landmarks in the GMM. To explain this, we have added the following sentence:

"To characterise vertebral morphology we used the combination of 15 morphological landmarks and qualitative characters shown previously to delineate morphological regions that are consistent with Hox gene expression limits in Gallus gallus domesticus (26) (Fig S1, Table S1)."

P5L116: it is written that the data were subjected to a Procrustes analysis for the PTA analysis. Does this mean there was a second Procrustes fit in addition to the one mentioned before? Please clarify.

No, the data were not subjected to a second Procrustes analysis for the PTA.

P6L120f: if I understand right the PTA based on connecting the mean of one region across ecologically similar birds with the mean of the next region. But each birds' regional shape already is the mean of the vertebrae within this region – basically resulting in the comparison of means of means. Is this really reasonable. Are these means of means still biological relevant. Wouldn't it be better to always took the middle vertebrae of each region and then use the mean of these representative vertebrae for region# across ecologically similar species?

Yes this does create means of means. We visually checked all examples to ensure that creating means of means did not create any artefacts or misrepresentations. This approach has been used in previous PTA analyses of vertebrae, and creating means of means is commonplace in other areas of shape analysis in biological research (e.g. plantar pressure and footprint analysis).

P7f149: scaling and ecology are labelled as extrinsic factors in this sentence. In the rest of the manuscript, however, size is used as an intrinsic factor. Please clarify.

We thank the reviewer for picking up on this. We have amended to read “*D-PGLS (in the R package ‘geomorph’ (48)) was used to assess the correlation between regional counts of cervical vertebrae and intrinsic (size) and extrinsic factors (diet, locomotion).*”

The Result section is quite long and as a reader one gets a little bit lost in details. The impact of extrinsic and intrinsic factors might be better presented in a table to show which factor best explains variability in which module. Morphological details and statistics could then be limited to the supplements and only most major trends could be summarized in the main text. As the journal usually publishes shorter manuscripts addressed to a wide readership, this could help to reduce length and to increase readability for researcher outside the field.

We thank the reviewer for this advice. We were attempting to be as thorough and robust as possible in describing our results, and our first submission did represent an extensive abbreviation of the total number of statistical models we ran in our analyses. However, we agree with the reviewer that sections (a) and (c) in the results section of our initial submission did contain too much anatomical detail and statistical description. We have therefore halved section (a), removing anatomical details. The previous (full) version of section (d) now exists in the supplementary information and we have replaced with an abbreviated version in the main text, which outlines only the major trends in terms of intrinsic and extrinsic factors influencing the different cervical regions. Section (d) is now almost half the length it was in the first submission.

As with the Results, the Discussion section losses itself a little bit in details. As the introduction missed to introduce details on previous finding on neck modularity in birds and mammals, it is hard for the reader to really asses what is a new finding and what is the impact of certain findings. The discussion needs a clear statement on which results agree with previous findings, which are contrary and which are new. In the next step, it would be useful to summarize in which features the avian neck is different to the mammalian neck (despite the variability in vertebral number). It is not sufficient just to state that birds are different. And in several instances, your findings even reveal similarities to mammals: neck elongation is primarily achieved by increasing midcervical region rather than upper or lower ones, mechanically demanding ecological behavior is associated with deviation from a more conserved ground plan etc. (Arnold et al. 2017 Evolution).

We thank the reviewer for this suggestion, but we feel quite strongly that the structure and layout of our discussion section is well suited to wider discussion of the results we present. We have

therefore retained the structure we used in our first submission. We start with an overview paragraph, which summarises the more important aspects of our data/results overall. It serves as an abstract for our discussion section. We have then broken our discussion down into three sections, each with a title that makes a statement that is a summation of the section's contents. We feel that this structure is comprehensible to the general biological readership and provides a good basis to discuss our key findings.

With regards to the comment above "*It is not sufficient just to state that birds are different.*" When we re-read our first submission it is not clear to us where we have simply stated that birds are different and then failed to discuss this in greater detail. We do this in our initial summary paragraph at the beginning of the discussion, but as noted above, the purpose of this paragraph is to pull out and summarise the most interesting findings from the results, which are then discussed in detail in the three subsections that follow. In these sub-sections, where we have found similar results to previous studies (e.g. isometrical scaling of neck length to body mass) we noted this and cited appropriately (in this submission and the previous one). Where our results contradict previous work we have noted this (e.g. vertebral elongation in birds driving increasing neck length, not addition of more vertebrae) and cited appropriately. We cannot see any parameter or factor analysed where we stated birds are different, and then not discussed it further.

With regards to the comparisons with Arnold et al. (2017), we would also emphasise again that as well as scaling patterns, we also show that the inter- and intra-regional morphology of vertebrae are modified by both 'extreme' and 'non-extreme' ecologies (not just extreme as the reviewer implies here), and that the pattern of change appears to be scale dependent. Arnold et al. (2017) only investigated region lengths in mammals (not the morphology of vertebrae and the nature of regionalisation as we do). We also explicitly discussed the relative pattern of elongation recovered for mammals by Arnold et al. (2017) versus our pattern for birds in our first submission. We stated:

"our data suggests that vertebral elongation is the primary mechanism by which neck elongation occurs, specifically it increases in the length of vertebrae in all regions except region 1 that are the epicentres of neck elongation across Aves (Table S5). This is in contrast to the more localised method by which mammalian neck elongation occurs, as it is vertebrae from just the middle portion of the neck length that lengthen (18)."

Where reference 18 was Arnold et al. (2017).

Moreover, some sections of the discussion are very speculative. Conservation in inter-regional morphology (P15L341ff) is speculated to be based on the optimization of the neck as a "hand" (i.e. optimizing motor pattern and control). As (almost) all mammals have only seven cervical vertebrae, this has been proposed as an explanation for conserved modular differences in vertebral morphology. However, such a functional explanation is hardly applicable to birds with varying number of cervical vertebrae. A conserved pattern of Hox gene expression might also be a good explanation.

In this paragraph we suggest that our results "*add credence*" to the hypothesis that at the gross level the avian neck "*is adapted to the 'economics of continuous movement' than to any specific ecology or behaviour (6,58,60,66).*" We maintain that this is perfectly rational argument to make, but we thank the reviewer for pointing out that alternative interpretations should be discussed, which we have done at the end of this paragraph.

Care should also be taken when explaining modifications related to carnivory. From a biomechanical point of view one would of course expect modifications in the lower cervical spine as even small changes in inter-vertebral mobility in this region would result in large changes in

angular excursion of the head. However, carnivory does not always equal carnivory – even among vultures there are big differences on how they obtain food from carrion which are reflected their neck morphology (Bohmer et al. 2020 J Anat). Therefore, the authors should explicitly refer to the limitations of such inferences.

Thank you for noting this point. We do feel it is worth briefly mentioning this in our manuscript and we have now done so. However, we do not feel that doing so weakens our analyses or highlights any limitations in our inferences about carnivory. Our results show that carnivory correlates with modifications to construction of the neck at various scales. That inference is not weakened by a finding that there is qualitative variation in neck musculature within carnivores. It is likely that all sub-groups have some level of variation or deeper specialisation within them, but it doesn't undermine differences between groups if they are recovered as statistically different.

Referee: 2

Comments to the Author(s)

The paper investigates the relationship between avian neck morphology and ecology such as feeding and locomotion. This is an area that is seeing interest given recent developments in detecting regionalization and increased interest in the axial skeleton. In general I believe the paper is strong and covers patterns which are not in the current literature.

We thank the reviewer for these positive comments, and particularly the recognition that our paper provides important findings that are not present in previous literature.

My largest issue with the paper as it stands is that in my opinion the discussion of neck length and head/body mass is too strongly simplistic. There is definitely a relationship between neck length and mass (Fig 3), but body/head mass are poor predictors of neck length, particularly when it is noted that the plots are log transformed. The confidence intervals for head mass look to maybe capture ~30% of the data. I expect biological data to be messy with low explanatory power, but since this is a comparison entirely within (an admittedly diverse) single clade, some discussion of other possible factors I think is reasonable. I agree that the data the authors gathered do not explain deviation from this relationship well, so what other factors might be contributing?

Thank you for highlighting this. We believe this is partly down to the style with which we presented and discussed these issues, but we also agree with the reviewer that more explicit discussion would be beneficial (note, we did briefly discuss this point in our supplementary information under “Additional Discussion”). The scaling results section (section d) was previously quite jumbled. We have reordered the relevant results section so that it now hopefully has the following structure or flow: (1) statements describing various scaling relationships, with note that there is quite a lot of variability; (2) then statements regarding the models that incorporate additional variables that might better explain the variation in the data. In the discussion we have renamed the heading of the subsection in which we discussed these data to highlight the variability explicitly (now called “*The unique nature of avian neck elongation, its constraints and variability*”). Then in this section we have attempted to discuss the variability and potential causes/correlations in a new paragraph, which includes specific factors (phylogeny, leg length) and the general consideration that the neck is an “all-purpose” organ and thus under a complex myriad of selective pressures. We welcome further suggestions if the reviewer feels we have not quite done this satisfactorily or if there are things we have not considered.

Following the Handbook of Avian Anatomy I recommend using cranial/caudal instead of anterior/posterior to eliminate possible confusion.

We thank the reader for this recommendation, which we have implemented in our resubmission.

100: Is this supposed to be 48? Or is the earlier number of 48 species incorrect? Or were masses not collected for every species?

The previous mention of 48 species referred to the analysis of neck length and morphology of vertebrae. Here we are referring to the analysis of head size and unfortunately we only had CT scans of the heads of 38 of the 48 birds, and so could only assess head size in this smaller number.

624: Missing reference information.

Thank you for pointing this out. This has been corrected.

Figure 2: Trajectories are hard to see on these small plots.

Thank you for pointing this out. In hindsight we agree completely. We have moved the legends outside the graphs, allowing expansion of the data to fill the graph. We've also now plotted the lines above/on top of the data points so they are clearer (rather than the lines being beneath the data points as previously).

Is Figure 3 referenced in the text?

Thank you for pointing this out. Figure 3 is now referenced at appropriate points in the results and discussion.

Appendix B

Response to Reviewer and Editor Comments

We are very grateful to the editor and reviewer for taking time to read our manuscript, and for the critical but thoughtful comments. The only required addition was to make our 3D bone models freely available, which we have done using Dyrad. The link is included in this submission.